# Antipsychotic Abuse, Dependence, and Withdrawal in the Pediatric Population: A Real-World Disproportionality Analysis

**DOI:** 10.3390/biomedicines10112972

**Published:** 2022-11-18

**Authors:** Diane Merino, Alexandre O. Gérard, Alexandre Destere, Florence Askenazy, Milou-Daniel Drici, Susanne Thümmler

**Affiliations:** 1Department of Child and Adolescent Psychiatry, Children’s Hospitals of Nice, CHU-Lenval, 06200 Nice, France; 2CoBTek Laboratory, Université Côte d’Azur, 06000 Nice, France; 3Pharmacovigilance Center, Department of Pharmacology, University Hospital of Nice, 06000 Nice, France

**Keywords:** psychotropic treatment, antipsychotics, misuse, addictology, adverse drug reaction, child and adolescent psychiatry

## Abstract

Antipsychotic drugs (APs) aim to treat schizophrenia, bipolar mania, and behavioral symptoms. In child psychiatry, despite limited evidence regarding their efficacy and safety, APs are increasingly subject to off-label use. Studies investigating addictology-related symptoms in young people being scarce, we aimed to characterize the different patterns of AP misuse and withdrawal in children and adolescents relying on the WHO pharmacovigilance database (VigiBase^®^, Uppsala Monitoring Centre, Sweden). Using the standardized MedDRA Query ‘drug abuse, dependence and withdrawal’, disproportionality for each AP was assessed with the reporting odds ratio and the information component. A signal was detected when the lower end of the 95% confidence interval of the information component was positive. Results revealed mainly withdrawal symptoms in infants (under 2 years), intentional misuse in children (2 to 11 years), and abuse in adolescents (12 to 17 years). Olanzapine, risperidone, aripiprazole, and quetiapine were disproportionately reported in all age groups, with quetiapine being subject to a specific abuse signal in adolescents. Thus, in adolescents, the evocation of possible recreational consumption may lead to addiction-appropriate care. Further, in young patients with a history of AP treatment, a careful anamnesis may allow one to identify misuse and its role in the case of new-onset symptoms.

## 1. Introduction

Since the discovery of chlorpromazine in France in the 1950s [1], antipsychotic drugs (APs) have aimed to relieve disorganized thoughts and behaviors, hallucinations, and delusions. Accordingly, they are used to treat schizophrenia, bipolar mania, and behavioral symptoms, inter alia. The common mechanism underlying the efficacy of most APs is considered to be the antagonism of brain dopamine D2 receptors [2]. In contrast with older drugs, referred to as ‘typical (or first-generation) antipsychotics’ (TAPs), the more recent ones (‘atypical (or second-generation) antipsychotics’, AAPs) are characterized by a stronger binding affinity to the serotonin 5HT2 receptor, compared to dopamine D2 receptors [3].

In child psychiatry, despite limited evidence regarding their efficacy and safety profiles, APs are increasingly subject to off-label use [4,5,6]. This phenomenon stems from a global increase in the prescribing of APs [7,8], a restricted pattern of marketing authorizations, as well as the lack of guidelines for their use in this population [9,10]. Aside from drug tolerance (which frequently leads to an increase in dosage), patients treated with APs are at risk of drug dependency [11,12,13]. Whether resulting from prescription or recreative purposes (such as seeking euphoria or relaxation), this consumption may lead to abuse, intentional misuse, but also withdrawal phenomena [14,15]. In younger populations, ‘pharming’, which involves the non-medical use and misuse of, mainly psychoactive, medication [16], may be favored in the growing role of the Internet and social media, especially regarding the easier accessibility of drugs [17,18]. Accordingly, the ‘psychonauts’ [14,16], ‘pharming’ users that try various psychoactive drugs and then share their experiences on social media, appeal to an ever-expanding audience, potentially influencing children and adolescents.

Studies investigating addictology-related symptoms in young people being scarce, beyond reports of accidental overdosage, we tried to characterize the different patterns of antipsychotic misuse and withdrawal, relying on an analysis of the World Health Organization (WHO) safety database (VigiBase^®^, Uppsala Monitoring Centre, Sweden) [19]. While withdrawal cases in infants and abuse cases in adolescents might be expected, there is still a grey area regarding middle-aged children. As antipsychotics are more and more subject to off-label use and illicit consumption in children and youth, we also aimed to identify potential drug safety signals regarding antipsychotic-related abuse, dependence, and withdrawal in this population. Further, we tried to shed some light on the most involved drugs, while suspecting the existence of different consumption patterns for each one, depending on the age group.

## 2. Materials and Methods

### 2.1. Data Source

The WHO mandates the Uppsala Monitoring Centre (UMC) to oversee drug safety [20]. This independent center aims to gather evidence about adverse drug reactions (ADRs), therefore leading to the identification of safety signals [21]. Indeed, VigiBase^®^ (UMC, Sweden), the WHO safety database, collects Individual Case Safety Reports issued by more than 172 national pharmacovigilance network members, along with pharmaceutical companies. The preservation of the anonymity of patients and notifiers is ensured by VigiBase^®^ (UMC, Sweden) [19].

Each individual Case Safety Report features sociodemographic characteristics of the patients (age, sex), administrative information (country, reporter qualification), suspected drug (indication, start and cessation dates, dose), concomitant drug(s), and characteristics of the ADR(s)’ (effect(s), seriousness, onset, outcome). In pharmacovigilance, an ADR is considered to be serious if it justified a hospitalization or its prolongation, caused a congenital malformation, resulted in persistent or significant disability or incapacity, was life threatening, resulted in death, or required significant medical intervention to prevent one of these outcomes [22,23].

### 2.2. Query

In the Medical Dictionary for Regulatory Activities (MedDRA, version 25.0 [24]), a Standardised MedDRA Query (SMQ) is an exhaustive, validated, predetermined collection of Preferred Terms (PTs) intended to help in investigating drug safety issues in pharmacovigilance [25]. A PT expresses a single medical concept in the most clinically accurate way [24].

We first queried VigiBase^®^ (UMC, Sweden) for all reports featuring the narrow SMQ ‘drug abuse, dependence and withdrawal’ (Medical Dictionary for Regulatory Activities, MedDRA 25.0 [24]) registered between 14 November 1967 (first reports in VigiBase^®^, (UMC, Sweden)) and 4 August 2022 and involving all antipsychotics reported in the database (acepromazine, acetophenazine, amisulpride, amperozide, aripiprazole, asenapine, benperidol, bifeprunox, blonanserin, brexpipirazole, bromperidol, butaperazine, carfenazine, cariprazine, chlorphenetazine, chlorproethazine, chlorpromazine, chlorprothixene, ciclofenazine, clopenthixol, clorotepine, clotiapine, clozapine, cyamemazine, dixyrazine, droperidol, fluanisone, flupentixol, fluphenazine, fluspirilene, haloperidol, iloperidone, lenperone, levomepromazine, levosulpiride, loxapine, lumateperone, lurasidone, melperone, mesoridazine, methopromazine, metofenazate, molindone, moperone, mosapramine, nemonapride, olanzapine, oxypertine, oxyprothepin, paliperidone, pecazine, penfluridol, perazine, periciazine, perospirone, perphenazine, pimavanserin, pimozide, pipamperone, piperacetazine, pipotiazine, pomaglumetad methionil, prochlorperazine, promazine, prothipendyl, quetiapine, raclopride, remoxipride, risperidone, sarizotan, sertindole, setoperone, sonepiprazole, spiperone, sulforidazine, sulpiride, sultopride, thiopropazate, thioproperazine, thioridazine, tiapride, timiperone, tiotixene, trifluoperazine, trifluperidol, triflupromazine, veralipride, ziprasidone, zotepine, and zuclopenthixol).

Then, queried reports involving patients aged under 18 were classified into 3 age groups: under 23 months, 2 to 11 years, and 12 to 17 years. For each group, records were aggregated depending on the PTs, into one 1 of 3 categories: misuse, abuse, or withdrawal [18].

‘Misuse’ can be defined as the intentional and inappropriate use of a product other than as prescribed or not in accordance with validated drug labels. It included the PTs ‘Intentional overdose’ and ‘Intentional product misuse’.

‘Abuse’ is the intentional non-therapeutic use of a product for the experience or feeling elicited (e.g., euphoria). It included the PTs ‘Drug abuse’, ‘Drug abuser’, ‘Drug dependence’, ‘Drug use disorder’, ‘Neonatal complications of substance abuse’, ‘Substance abuse’, ‘Substance abuser’, ‘Substance dependence’, and ‘Substance use disorder’.

‘Withdrawal’ describes symptoms or signs related to the cessation of a drug, either physiological withdrawal reaction to the drug or exacerbation of the underlying disease itself [26]. It included the PTs ’Drug withdrawal convulsions’, ‘Drug withdrawal headache’, ’Drug withdrawal syndrome’, and ‘Drug withdrawal syndrome neonatal’.

### 2.3. Statistical Analyses

Quantitative variables were described in terms of means with standard deviations (±SD). Qualitative variables were described with proportions. Statistical analyses were performed using GraphPad Prism version 8.0.2. Thereupon, we performed a disproportionality analysis, to partially mitigate the impact of potential confounding factors [27]. Potential pharmacovigilance signals were sought for all antipsychotics (as described above, see Section 2.2. Query), for each age group:Using the narrow SMQ ‘Drug abuse, dependence, and withdrawal’ when the chosen drug accounted for ≥2 cases and >1% of the reports of the respective age group;Using the most represented combination of PTs (in each age group), when the chosen drug accounted for ≥10% of the reports and/or was quetiapine, olanzapine, risperidone, and aripiprazole. This particular analysis aimed to clarify the specific ADRs driving any possible signal arising from the disproportionality analysis of the SMQ.

The disproportionality analysis relied on the reporting odds ratio (ROR) and the information component (IC).

As an approximate of the odds ratio (used in case-control studies), the ROR is estimated, in case–non-case studies, to assess the strength of disproportionality. An ROR equal to 1 indicates the absence of signal: the ADR is equally reported with the drug of interest as with the other drugs. In contrast, an ROR greater than 1 suggests the existence of a signal, as cases appear to be more frequently reported with the drug of interest than with the others. The higher the ROR, the stronger the association. A 95% confidence interval (95% CI) reflects the precision of the approximate ROR. Therefore, an ROR is deemed to be statistically significant when the lower bound of its 95% CI is greater than 1 [28].

The IC allows one to compare observed and expected values for the combination of a given drug and an ADR, to check for a potential association. It favors the reduction in the risk of false-positive signals, especially if the chosen ADR has a very low expected frequency in the database (therefore, mechanically increasing the ROR). The positivity of the IC reflects that the number of observed reports is higher than expected. The IC025, which is the bottom end of the 95% CI of the information component, is required to statistically confirm the detection of a signal in VigiBase^®^ (UMC, Sweden) [19,29].

In this disproportionality analysis, potential drug–ADR associations were selected by using IC025. Then, the ROR of each drug–effect association allowed us to assess the strength of respective suggested signals. For this purpose, we used Microsoft^®^ Excel^®^ 2019 Version 2210.

## 3. Results

### 3.1. Drug Abuse, Dependence, and Withdrawal

As of 4 August 2022, 16,054 reports belonging to the narrow SMQ ‘Drug abuse, dependence and withdrawal’ and involving consumers of antipsychotics were collected in VigiBase^®^ (UMC, Sweden). Among these reports, 1023 (6.4%) involved patients below 18 years of age, mostly belonging to the 12-to-17-year group (732, 71.6%). Records with patients aged under 18 mostly originated from the United States (398, 38.9%). Healthcare professionals issued 84.8% of the cases, with a majority of physicians (617, 60.3%). Details regarding the characteristics of the reports are provided in Table 1.

Four antipsychotics accounted for 839 (82.0%) reports of abuse, dependence, and withdrawal: quetiapine (368, 36.0%), risperidone (224, 21.9%), aripiprazole (129, 12.6%), and olanzapine (118, 11.5%). The number of reports for all ADRs in patients aged under 18 involving each of these antipsychotics is displayed in Appendix A.

Seriousness was assessed in 925 (90.4%) reports: 846 (91.5%) were deemed serious, among which 488 (57.6%) ADRs caused/prolonged hospitalizations, 78 (9.2%) life-threatening reactions, 40 (4.7%) deaths, and 30 (3.5%) congenital anomalies/birth defects. When details regarding the outcome were available (429, 41.6%), 356 (83.0%) patients recovered or were recovering, 40 (9.3%) did not recover, and 6 (1.4%) recovered with sequelae.

In patients aged under 18, antipsychotics were subject to a disproportionate reporting for the SMQ ‘Drug abuse, dependence and withdrawal’ (ROR 5.5; IC025 2.2).

### 3.2. Patients Aged between 0 Days and 23 Months

Among patients below 18 years of age, 198 (19.4%) reports of ADRs related to antipsychotic abuse, dependence, and withdrawal and involving infants aged from 0 days to 23 months were found in VigiBase^®^ (UMC, Sweden) In this age group, males represented 51.5% (n = 102) and their mean age was 2.3 (±2.9) months. Most of them (146, 73.7%) were newborns (aged under 28 days). The most frequently co-reported MedDRA terms were fetal exposure during pregnancy (63, 31.8%), premature baby (27, 12.1%), and tremor (17, 8.6%). Quetiapine was suspected in 75 records (37.8%), followed by aripiprazole (30, 15.2%), risperidone (28, 14.1%), and olanzapine (20, 10.1%). The complete list of involved antipsychotics is provided in Appendix A.

Co-reported drugs were found in 177 cases (89.4%). The most frequently suspected co-reported active ingredients were other psychotropic drugs, such as antidepressants sertraline (20, 10.1%), clomipramine and fluoxetine (18 cases each, 9.1%), venlafaxine (17, 8.6%), and the hypnotic drug zopiclone (14, 7.1%).

When available, 140 (96.5%) reports were deemed serious, among which 89 (63.6%) ADRs caused/prolonged hospitalizations, 30 (21.4%) congenital anomalies/birth defects, 13 (9.3%) life-threatening reactions, and 1 (0.7%) death. Among reports with available follow-up, 145 (91.8%) infants recovered or were recovering, 10 (6.3%) were not recovering, and 2 (1.3%) recovered with sequelae.

The most often involved antipsychotics (absolute number of reports) were disproportionately reported with the SMQ ‘Drug abuse, dependence, and withdrawal’: quetiapine (ROR 68.3; 95% CI: 53.2–87.8), risperidone (ROR 35.2; 95% CI: 23.8–52.0), aripiprazole (ROR 25.1; 95% CI: 17.2–36.4), and olanzapine (ROR 23.4; 95% CI: 14.9–36.9). Apart from quetiapine, the highest RORs were reached by cyamemazine (ROR 82.1; 95% CI: 47.5–141.8), amisulpride (ROR 68.3; 95% CI: 23.3–200.1), zuclopenthixol (ROR 60.3; 95% CI: 17.6–205.8), and levomepromazine (ROR 58.2; 95% CI: 28.6–118.0). The whole disproportionality analysis for cases of infants aged between 0 and 23 months is displayed in Table 2. Figure 1 summarizes the disproportionality analysis for the main antipsychotics involved in reports of abuse, dependence, or withdrawal, depending on the age range.

Upon closer inspection, in the narrow SMQ ‘Drug abuse, dependence, and withdrawal’, most reports involving infants aged between 0 days and 23 months referred to PTs related to withdrawal (192, 96.7%), as displayed in Figure 2 and Appendix A.

The combination of ‘withdrawal’ PTs (drug withdrawal headache, drug withdrawal syndrome, drug withdrawal syndrome neonatal) showed disproportionate reporting for quetiapine (ROR 86.8; 95% CI: 67.4–111.8), risperidone (ROR 43.3; 95% CI: 29.1–64.5), aripiprazole (ROR 28.7; 95% CI: 19.4–42.5), and olanzapine (ROR 28.4; 95% CI: 17.8–45.3), as shown in Table 3.

### 3.3. Patients Aged between 2 and 11 Years

Children aged between 2 and 11 years accounted for 93 (9.1%) reports involving patients aged under 18 years. The majority of patients in this age group were male (65, 69.9%), with a mean age of 8.4 years (±2.2). Dyskinesia (16, 17.2%), dystonia (13, 14.0%), and aggression (11, 11.8%) were the most frequently co-reported MedDRA terms. In this age range, risperidone was the most frequently suspected antipsychotic (42, 45.2%), followed by quetiapine (23, 24.7%), aripiprazole 13 (13, 14.0%), and olanzapine (10, 10.8%). The complete list is available in Appendix A.

Co-reported drugs were found in 63 cases (67.7%). The most frequently suspected co-reported active ingredients were methylphenidate (10, 10.8%), levothyroxine (6, 6.5%), citalopram (6, 6.5%), valproic acid (5, 5.4%), and omeprazole (5, 5.4%).

When available, 61 cases (79.2%) were considered to be serious, with 36 (46.8%) causing/prolonging hospitalizations, 8 (10.4%) deaths, and 4 (5.2%) life-threatening reactions. Among reports with follow-up, 19 (63.3%) infants recovered or were recovering, 4 (13.3%) were not recovering, and 2 (6.7%) recovered with sequelae.

Quetiapine (ROR 19.3; 95% CI: 12.7–29.4), olanzapine (ROR 10.0; 95% CI: 5.3–18.7), risperidone (ROR 5.0; 95% CI: 3.7–6.3), and aripiprazole (ROR 3.1; 95% CI: 1.8–5.3) were subject to a disproportionate reporting with the SMQ ‘Drug abuse, dependence, and withdrawal’ (Table 4, Figure 1).

In the SMQ ‘Drug abuse, dependence and withdrawal’, most reports involving children aged between 2 and 11 years referred to PTs related to intentional misuse (46, 49.5%) and withdrawal (37, 39.8%) (Figure 2, Appendix A).

The ‘intentional misuse’ combination of PTs (intentional overdose, intentional product misuse) showed disproportionate reporting for quetiapine (ROR 34.4; 95% CI: 22.0–54.0), followed by olanzapine (ROR 12.2; 95% CI: 5.4–27.2) and risperidone (ROR 4.3; 95% CI: 2.7–6.9) (Table 5). No report was found for aripiprazole in this category. The median time to onset for intentional misuse-related ADRs was 730 days (IQR 122–1460).

The ‘withdrawal’ combination of PTs (drug withdrawal syndrome, drug withdrawal convulsions) was subject to disproportionate reporting for olanzapine (ROR 13.7; 95% CI: 5.1–36.7), aripiprazole (ROR 10.0; 95% CI: 5.6–17.7), and risperidone (ROR 7.8; 95% CI: 4.9–12.4) (Table 6).

### 3.4. Patients Aged between 12 and 17 Years

Adolescents aged between 12 and 17 years accounted for 732 (71.6%) of the queried reports. In this age group, most patients were female (408, 55.7%) and their mean age was 15.3 (±1.5) years, with more than one-quarter being 17-year-old adolescents (199, 27.2%). The most frequently co-reported MedDRA terms were suicide attempt (226, 30.9%), somnolence (156, 21.3%), and tachycardia (62, 8.5%). Quetiapine accounted for 270 records (36.9%), followed by risperidone (154, 21.0%), olanzapine (88, 12.0%), and aripiprazole (86, 11.7%). The complete list of suspected antipsychotics for this age range is available in Appendix A.

Other drugs were co-reported in 537 cases (73.4%). Fluoxetine (54, 7.4%), alprazolam (39, 5.3%), sertraline (36, 4.9%), paracetamol (35, 4.8%), and diazepam (28, 3.8%) were the most frequently suspected co-reported active ingredients.

When this information was available, 645 (61.7%) reports were deemed serious, including 363 (51.6%) causing/prolonging hospitalizations, 61 (8.6%) life-threatening reactions, and 31 (4.4%) deaths. Among reports with follow-up, 192 (76.7%) infants recovered or were recovering, 26 (10.7%) were not recovering, and 2 (0.8%) recovered with sequelae.

The most frequently involved antipsychotics were disproportionately reported with the narrow SMQ ‘Drug abuse, dependence and withdrawal’: quetiapine (ROR 8.9; 95% CI: 7.8–10.1), olanzapine (ROR 2.8; 95% CI: 2.3–3.5), risperidone (ROR 1.8; 95% CI: 1.5–2.1), and aripiprazole (ROR 1.7; 95% CI: 1.4–2.1) (Table 7, Figure 1). Beyond quetiapine, promazine (ROR 97.5; 95% CI: 56.0–169.8), chlorprothixene (ROR 35.5; 95% CI: 22.1–57.3), pipamperone (ROR 14.6; 95% CI: 8.5–25.0), and cyamemazine (ROR 4.2; 95% CI: 2.6–6.6) showed the greatest RORs.

Most of these reports referred to PTs related to intentional misuse (545, 74.5%) and abuse (168, 23.0%), as displayed in Figure 2 and Appendix A.

The ‘intentional misuse’ combination of PTs (intentional overdose, intentional product misuse) was subject to disproportionate reporting for quetiapine (ROR 10.7; 95% CI: 9.3–12.3), olanzapine (ROR 3.6; 95% CI: 2.8–4.6), risperidone (ROR 2.1; 95% CI: 1.7–2.5), and aripiprazole (ROR 1.9; 95% CI: 1.5–2.5) (Table 8).

The ‘abuse’ combination of PTs (drug abuse, drug dependence, substance abuse, drug abuser, drug use disorder, substance abuser, substance dependence, substance use disorder) was only disproportionately reported for quetiapine (ROR 4.8; 95% CI: 3.7–6.3), (Table 9).

## 4. Discussion

Our analysis of the WHO pharmacovigilance database brings to light varying profiles for abuse, dependence, and withdrawal related to antipsychotics. Indeed, in infants aged under 2 years, almost all ADRs were understandably related to withdrawal symptoms. In children aged between 2 and 11 years, intentional misuse was at the forefront, with fewer cases of withdrawal, frequently complicated by extrapyramidal symptoms. Lastly, adolescents aged from 12 to 17 years were subject to intentional misuse, but also to abuse issues.

In infants aged from 0 days to 23 months, withdrawal syndrome reflects prenatal maternal exposure to APs. AAPs (quetiapine, risperidone, aripiprazole, olanzapine) prevailed, both in terms of absolute number of cases and disproportionality, which is consistent with existing literature regarding their respective safety profiles and prescribing trends during pregnancy [30,31]. However, their drug labels indicate a risk of withdrawal in exposed neonates and recommend avoiding these drugs during pregnancy, unless their use is an absolute necessity [32,33,34,35]. Quetiapine showed the strongest signal in this population. Indeed, in women previously stabilized with quetiapine, it often remains prescribed during pregnancy [36,37]. At birth, the newborn being severed from the maternal blood supply and quetiapine being rapidly dissociated from the D2 receptors [38,39], its abrupt discontinuation may lead to greater risks of withdrawal at birth [40]. The frequent maternal cotreatment with antidepressants (e.g., mood disorders) that we observed may also reinforce withdrawal symptoms in infants [40,41,42].

In children aged between 2 and 11 years, the age and gender distribution regarding treatment with antipsychotics in the literature concurs with our findings [43,44,45]. Indeed, significant disproportionality was found for AAP only, subject to a dramatic increase in consumption in children, circumscribed by the rise of off-label use in this population [43,44,46,47]. Regarding risperidone, which is the leading drug in terms of absolute number of reports, both the Food and Drug Administration (FDA) and the European Medicines Agency (EMA) have granted it marketing authorization in irritability occurring in patients with an autistic disorder from the age of 5 [35,48]. Aripiprazole is authorized by the FDA from the age of 6 [34] in the same indication, but its use is only recommended from the age of 13 by the EMA (manic episodes of bipolar I disorder) [49]. Lastly, marketing authorization for olanzapine and for quetiapine is granted by the FDA in acute mixed or manic episodes of bipolar I disorder from the age of 10 [32,33]. However, the EMA stated that their safety and efficacy in children under the age of 18 have not been established yet and, therefore, that they should not be used in this group until further data become available [50,51]. In more than one-quarter of the reports, abuse, dependence, or withdrawal occurred concomitantly with dystonia and/or dyskinesia. These effects may reflect either overdosing following intentional misuse or withdrawal syndrome [52,53], subsequent to dopamine receptor hypersensitivity, GABA insufficiency, or cellular degeneration (neurotoxicity) [54,55]. In cases of withdrawal, the strongest disproportionality signals concerned risperidone and aripiprazole. This may indicate abrupt treatment discontinuations that could be favored by the urge of appeasing irritability and/or aggressive symptoms via the initiation of a new medication (e.g., in patients suffering from autistic disorders). In addition, possible impulsive and voluntary intoxications may also occur, especially in children presenting with behavioral disorders.

Adolescents aged from 12 to 17 years accounted for the vast majority of the reports belonging to the SMQ ‘drug abuse, dependence and withdrawal’. Paracetamol was quite frequently co-reported as suspect or interacting, which may reflect the fact that suicide attempts accounted for nearly one-third of the reports in this age range [56]. Further, ingestion of co-reported substances, such as antidepressants and benzodiazepines, might also concur with phenomena of intentional misuse or abuse or designate a population of patients receiving long-term treatment with psychotropic drugs. In this context, the AAPs quetiapine, olanzapine, risperidone, and aripiprazole were subject to disproportionate reporting, to a greater degree than TAPs, such as promazine, chlorprothixene, and cyamemazine. Regarding TAPs, cyamemazine has been granted marketing authorization in France (Agence Nationale du Médicament et des produits de santé—ANSM) for behavioral disorders with psychomotor agitation and aggressivity for children aged 3 years or older [57]. Promazine and chlorprothixene were withdrawn from the market in the United States [58,59]. However, commonly prescribed AAPs may be ingested for recreative purposes, as Novel Psychoactive Substances (NPSs) [12,18]. According to the United Nations Office on Drugs and Crime (UNODC), NPSs are ‘substances of abuse, either in a pure form or a preparation, that are not controlled by the 1961 Single Convention on Narcotic Drugs or the 1961 Convention on Psychotropic Substances, but which may pose a public health threat’ [60]. Compared to ‘conventional’ illicit substances, NPSs are distinguished by greater affordability and ease of online purchase, but also lower detectability and social stigma [61]. In younger populations, both appeasement of other consumption symptoms and search of emotional anesthesia are becoming more and more accessible, as these phenomena are further exacerbated by the ‘pharming’ and ‘psychonauts’ trends. As abuse and intentional misuse have been described for risperidone and aripiprazole [13,62], we extend this potential signal to adolescent populations. In line with previous findings (irrespective of the age range) [13,15], olanzapine was disproportionately reported for intentional misuse. Olanzapine can be consumed as a psychedelic or for its sedative effect, with rewarding properties involving glutamatergic stimulation of dopaminergic neurons of the ventral tegmental area [63]. Our findings confirm that quetiapine, already involved in pharmacovigilance signals of intentional misuse and abuse, raises also issues in adolescents. Indeed, it is the leading drug in terms of absolute number of reports. Quetiapine has multiple street names, such as ‘quell’, ‘Susie-Q’, ‘baby heroin’, and ‘Q-ball’ (when associated with cocaine) [64]. Consumed for anxiolytic properties, its popularity as an NPS may be linked to H1 and α1 receptor antagonism. The same rationale may apply to olanzapine, more and more available on the black market [13,64,65].

While other studies aimed to assess the importance of antipsychotic abuse and misuse in different pharmacovigilance databases [13,15,66], our analysis was the first to focus on children and youth. It highlighted different profiles, depending on the age of the patients. Nevertheless, this study is hindered by the inherent flaws of spontaneous reporting systems and post-marketing pharmacovigilance approaches, such as incomplete data and reporting bias. The strict definitions of the terms, from a pharmacovigilance perspective, might be unclear for notifiers, which may have led to a substantial coding heterogeneity, especially regarding intentional misuse and abuse. However, most reports were notified by healthcare professionals, which limits the risk of coding errors. Further, lack of follow-up and under-reporting, although usual in pharmacovigilance [67], may have been heightened by the extent of overall consumption and the acute nature of manifestations (especially related to withdrawal or intentional overdose). Then, associated medications were recorded (co-reported active ingredients), but we were not able to retrieve data on the consumption of other illegal substances, as this is outside the scope of VigiBase^®^(UMC, Sweden) Lastly, pharmaco-epidemiological studies aim to raise awareness about possible drug safety signals and no definite causality can be drawn from our findings.

Regarding implications for practice and research, different areas have to be taken into account. During pregnancy, as recommended by the safety agencies [32,33,34,35,48,49,50,51], each antipsychotic treatment must be introduced or maintained following a careful assessment of the benefit/risk ratio. In children, off-label prescription should be properly substantiated, considering the scarcity of data regarding safety in efficacy in this population [50,51]. In adolescents, the risk of antipsychotic abuse and its characteristics should be further investigated. In addition, particular heed should be paid to specialized social networks, underlying new consumption patterns [14,16].

## 5. Conclusions

In this study, relying on a comprehensive analysis of drug abuse, dependence, and withdrawal reports with APs from the WHO safety database, we suggest potential pharmacovigilance signals involving several AAPs, especially quetiapine, olanzapine, risperidone, and aripiprazole. Defining three profiles according to the patients’ age ranges, we highlighted the potential role of these drugs in withdrawal phenomena among infants, but also in intentional misuse behaviors among children and adolescents. We confirmed existing signals of intentional misuse of APs and the growing importance of quetiapine as an NPS. In young patients with a history of AP treatment, a careful anamnesis may allow one to identify the role of these medications in the case of new-onset symptoms. Indeed, given the variety in their manifestations, the hypothesis of an antipsychotic-related ADR should be systematically evoked. In adolescents, the evocation of a possible recreational consumption may lead to addiction-appropriate care, therefore, reducing further morbidity.

## Figures and Tables

**Figure 1 biomedicines-10-02972-f001:**
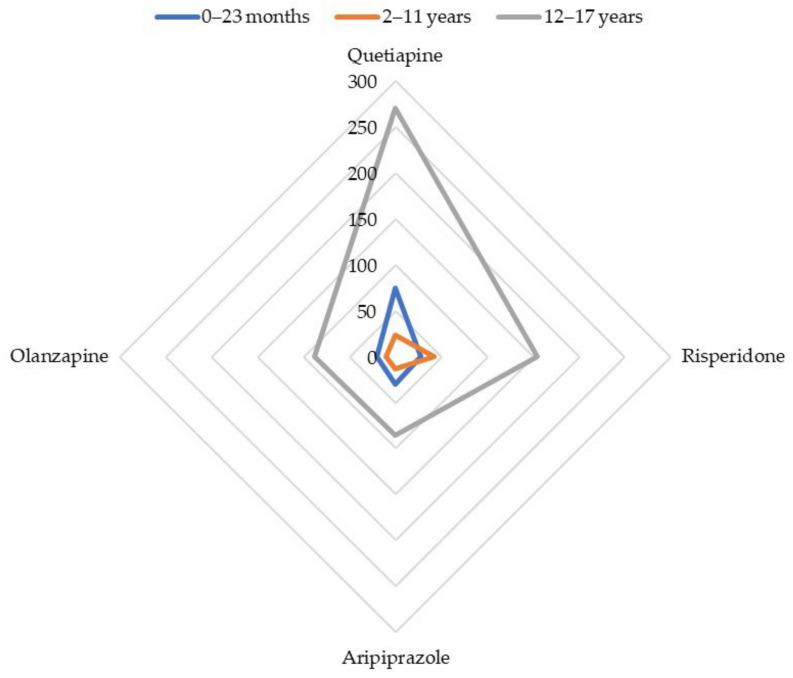
Cases’ distribution for main antipsychotics involved in reports of abuse, dependence, or withdrawal.

**Figure 2 biomedicines-10-02972-f002:**
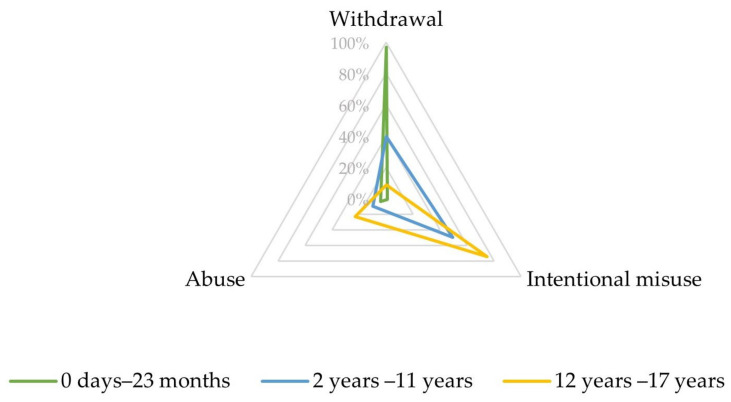
Characteristics of the combined Preferred Terms (PTs) ‘abuse’, ‘intentional misuse’, and ‘withdrawal’ depending on age.

**Figure 3 biomedicines-10-02972-f003:**
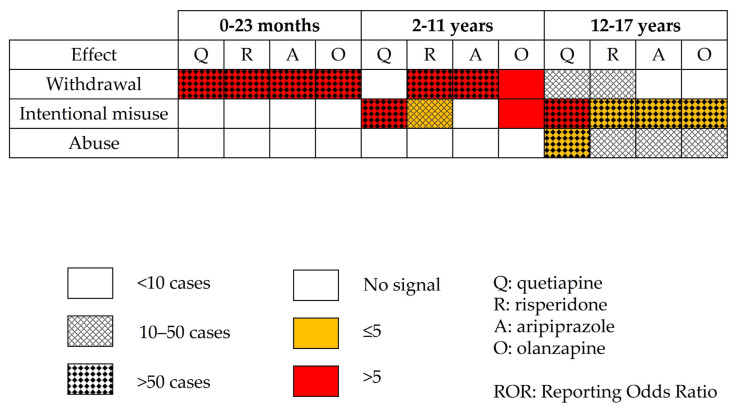
Disproportionality analysis using combined Preferred Terms for most involved antipsychotics.

**Table 1 biomedicines-10-02972-t001:** Characteristics of the reports of patients with abuse, dependence, or withdrawal.

Characteristics	Number of Reports (%)			
	0 Days–23 Months	2 Years–11 Years	12 Years–17 Years	Total <18 Years
Total:	198 (100)	93 (100)	732 (100)	1023 (100)
**Sex**				
Female	74 (37.4)	26 (28.0)	408 (55.7)	508 (49.7)
Male	102 (51.5)	65 (69.9)	291 (39.8)	458 (44.8)
Unknown	22 (11.1)	2 (2.2)	33 (4.5)	57 (5.6)
**Country**				
United States of America	26 (13.1)	58 (62.4)	314 (42.9)	398 (38.9)
Germany	32 (16.6)	1 (1.1)	103 (14.1)	136 (13.3)
Croatia		1 (1.1)	92 (12.6)	93 (9.1)
France	44 (22.2)	2 (2.2)	35 (4.8)	81 (7.9)
Italy		2 (2.2)	75 (10.2)	77 (7.5)
Australia	44 (22.2)	2 (2.2)	11 (1.5)	57 (5.6)
United Kingdom	9 (4.5)	5 (5.4)	22 (3.0)	36 (3.5)
Canada	9 (4.5)	9 (9.7)	8 (1.1)	26 (2.5)
Japan	23 (11.6)		3 (0.4)	26 (2.5)
Austria			16 (2.2)	16 (2.2)
Switzerland	4 (2.0)		9 (1.2)	13 (1.3)
Sweden			11 (1.5)	11 (1.1)
Spain		6 (6.5)	1 (0.1)	7 (0.7)
Turkey			6 (0.8)	6 (0.6)
Brazil			5 (0.7)	5 (0.5)
Finland	3 (1.5)		1 (0.1)	4 (0.4)
Portugal			4 (0.5)	4 (0.4)
Denmark	3 (1.5)			3 (0.3)
India		2 (2.2)	1 (0.1)	3 (0.3)
Netherlands		1 (1.1)	2 (0.3)	3 (0.3)
Poland		1 (1.1)	2 (0.3)	3 (0.3)
South Africa		1 (1.1)	2 (0.3)	3 (0.3)
Bulgaria			2 (0.3)	2 (0.2)
Korea	1 (0.5)		1 (0.1)	2 (0.2)
Lithuania		1 (1.1)	1 (0.1)	2 (0.2)
Norway			2 (0.2)	2 (0.2)
Belgium			1 (0.1)	1 (0.1)
Eritrea		1 (1.1)		1 (0.1)
Ireland			1 (0.1)	1 (0.1)
New Zealand			1 (0.1)	1 (0.1)
**Reporter qualification**				
**Healthcare Professional**	**165 (83.3)**	**67 (72.0)**	**636 (86.9)**	**868 (84.8)**
Physician	140 (70.7)	45 (48.4)	432 (59.0)	617 (60.3)
Pharmacist	8 (4.0)	6 (6.5)	42 (5.7)	56 (5.5)
Other Health Professional	17 (8.6)	16 (17.2)	162 (22.1)	195 (19.1)
**Others**	**32 (16.2)**	**17 (18.3)**	**86 (11.8)**	**135 (13.2)**
Lawyer	1 (0.5)		2 (0.3)	3 (0.3)
Consumer	31 (15.7)	17 (18.3)	84 (11.5)	132 (12.9)
**Unknown**	**18 (9.1)**	**11 (11.8)**	**41 (5.6)**	**70 (6.8)**

**Table 2 biomedicines-10-02972-t002:** Disproportionality analysis for reports of abuse, dependence, or withdrawal in patients aged between 0 days and 23 months, ranked by ROR.

Antipsychotic	IC025	ROR	95% CI	Number (%)
All antipsychotics	4.6	35.0	30.1–40.8	198 (100)
Cyamemazine	3.7	82.1	47.5–141.8	16 (8.1)
Quetiapine	5.0	68.3	53.2–87.8	75 (37.8)
Amisulpride	1.2	68.3	23.3–200.1	4 (2.0)
Zuclopenthixol	0.6	60.3	17.6–205.8	3 (1.5)
Levomepromazine	2.7	58.2	28.6–118.0	9 (4.5)
Chlorpromazine	3.3	35.9	21.7–59.2	17 (8.6)
Risperidone	3.8	35.2	23.8–52.0	28 (14.1)
Aripiprazole	3.5	25.1	17.2–36.4	30 (15.2)
Olanzapine	3.2	23.4	14.9–36.9	20 (10.1)
Haloperidol	2.0	19.0	9.3–38.8	8 (4.0)
Clozapine	1.9	15.7	7.7–32.0	8 (4.0)

ROR: Reporting Odds Ratio; IC: Information Component; CI: Confidence Interval.

**Table 3 biomedicines-10-02972-t003:** Disproportionality analysis for main antipsychotics involved in reports of withdrawal in patients aged between 0 days and 23 months, ranked by ROR.

Antipsychotic	IC025	ROR	95% CI	Number (%)
Quetiapine	5.2	86.8	67.4–111.8	74 (39.2)
Risperidone	3.9	43.3	29.1–64.5	27 (14.3)
Aripiprazole	3.6	28.7	19.4–42.5	27 (14.3)
Olanzapine	3.3	28.4	17.8–45.3	19 (10.1)

**Table 4 biomedicines-10-02972-t004:** Disproportionality analysis for reports of abuse, dependence, or withdrawal in patients aged between 2 years and 11 years, ranked by ROR.

Antipsychotic	IC025	ROR	95% CI	Number (%)
All antipsychotics	2.0	5.2	4.2–6.4	93 (100)
Quetiapine	3.1	19.3	12.7–29.4	23 (24.7)
Olanzapine	1.8	10.0	5.3–18.7	10 (10.8)
Risperidone	1.7	5.0	3.7–6.3	42 (45.2)
Aripiprazole	0.6	3.1	1.8–5.3	13 (14.0)
Paliperidone	−0.3	5.4	1.8–17.0	3 (3.2)
Cyamemazine	−0.7	11.4	2.8–46.4	2 (2.2)

**Table 5 biomedicines-10-02972-t005:** Disproportionality analysis for main antipsychotics involved in reports of intentional misuse in patients aged between 2 years and 11 years, ranked by ROR.

Antipsychotic	IC025	ROR	95% CI	Number (%)
Quetiapine	3.5	34.4	22.0–54.0	20 (43.5)
Olanzapine	1.3	12.2	5.4–27.2	6 (13.0)
Risperidone	1.2	4.3	2.7–6.9	18 (39.1)
Aripiprazole				0

**Table 6 biomedicines-10-02972-t006:** Disproportionality analysis for main antipsychotics involved in reports of withdrawal in patients aged between 2 years and 11 years, ranked by ROR.

Antipsychotic	IC025	ROR	95% CI	Number (%)
Olanzapine	0.8	13.7	5.1–36.7	4 (10.8)
Aripiprazole	1.9	10.0	5.6–17.7	12 (32.4)
Risperidone	2.0	7.8	4.9–12.4	19 (51.4)
Quetiapine	−3.0	2.8	0.39–19.7	1 (2.7)

**Table 7 biomedicines-10-02972-t007:** Disproportionality analysis for reports of abuse, dependence, or withdrawal in patients aged between 12 years and 17 years, ranked by ROR.

Antipsychotic	IC025	ROR	95% CI	Number (%)	
All antipsychotics	1.1	2.5	2.3–2.7	732 (100)	
Promazine	4.1	97.5	56.0–169.8	29 (4.0)	
Chlorprothixene	3.4	35.5	22.1–57.3	25 (3.4)	
Pipamperone	2.4	14.6	8.5–25.0	16 (2.2)	
Quetiapine	2.8	8.9	7.8–10.1	270 (36.9)	
Cyamemazine	1.2	4.2	2.6–6.6	19 (2.6)	
Olanzapine	1.1	2.8	2.3–3.5	88 (12.0)	
Amisulpride	0.1	2.8	1.4–5.6	8 (1.1)	
Risperidone	0.5	1.8	1.5–2.1	154 (21.0)	
Aripiprazole	0.4	1.7	1.4–2.1	86 (11.7)	
Ziprasidone	0.1	1.0	1.2–3.3	15 (2.0)	
Clozapine	−0.8	0.9	0.6–1.2	29 (4.0)	
Haloperidol	−0.7	1.0	0.6–1.5	21 (2.9)	

**Table 8 biomedicines-10-02972-t008:** Disproportionality analysis for main antipsychotics involved in reports of intentional misuse in patients aged between 12 years and 17 years, ranked by ROR.

Antipsychotic	IC025	ROR	95% CI	Number (%)
Quetiapine	3.0	10.7	9.3–12.3	206 (37.8)
Olanzapine	1.4	3.6	2.8–4.6	70 (12.8)
Risperidone	0.7	2.1	1.7–2.5	114 (20.9)
Aripiprazole	0.5	1.9	1.5–2.5	62 (11.4)

**Table 9 biomedicines-10-02972-t009:** Disproportionality analysis for main antipsychotics involved in reports of abuse in patients aged between 12 years and 17 years, ranked by ROR.

**Antipsychotic**	**IC025**	**ROR**	**95% CI**	**Number (%)**
Quetiapine	1.8	4.8	3.7–6.3	55 (33.3)
Risperidone	−0.9	0.8	0.6–1.2	25 (15.1)
Aripiprazole	−0.7	1.0	0.6–1.6	18 (10.9)
Olanzapine	−1.2	0.9	0.5–1.7	10 (6.1)

Our findings are summarized in Figure 3.

## Data Availability

The data that support the findings of this study are available from Uppsala Monitoring Center (UMC) but restrictions apply to the availability of these data, which were used under license for the current study and so are not publicly available. Access to VigiBase^®^ is available without fees to Fanny Rocher. Data are, however, available from the authors upon reasonable request and with permission of UMC.

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
