# Peer review of "Antipsychotic Abuse, Dependence, and Withdrawal in the Pediatric Population: A Real-World Disproportionality Analysis"

_biomedicines, 2022, doi:10.3390/biomedicines10112972_

Round 1

Reviewer 1 Report

Well written and interesting.

Author Response

We thank Reviewer 1 for his/her appraisal of our manuscript.

Reviewer 2 Report

Thank you for giving me the opportunity to read and comment a report “Antipsychotic Abuse, Dependence, and Withdrawal in the Pediatric population: A Real-World Disproportionality Analysis”, by Merino D., et al.

This paper is well written, correctly structured with a suitable research concept, the study limitations are addressed, and it is of relevance to readers of the journal.

However, I include a few comments for your consideration.

·      It would be desirable for the authors to provide more detail on the main aim of the study.

·  Which statistical software did the authors use to perform the disproportionality analysis? If it is the same as for quantitative and qualitative variables, it would be convenient to indicate it. Perhaps it would be preferable to include the subsection "Statistical Analysis" in the "Materials and Methods" section.

·      The figures caption should be placed below the figure.

Author Response

Thank you for giving me the opportunity to read and comment a report “Antipsychotic Abuse, Dependence, and Withdrawal in the Pediatric population: A Real-World Disproportionality Analysis”, by Merino D., et al.

This paper is well written, correctly structured with a suitable research concept, the study limitations are addressed, and it is of relevance to readers of the journal.

We thank Reviewer 2 his/her interest in our manuscript.

However, I include a few comments for your consideration.

  •     It would be desirable for the authors to provide more detail on the main aim of the study.

As suggested by Reviewer 2, we added more detail on the main aim of the study: ‘Further, we tried to shed some light on the most involved drugs, while suspecting the existence of different consumption patterns for each one, depending on the age group.’

  • Which statistical software did the authors use to perform the disproportionality analysis? If it is the same as for quantitative and qualitative variables, it would be convenient to indicate it. Perhaps it would be preferable to include the subsection "Statistical Analysis" in the "Materials and Methods" section.

We thank Reviewer 2 for his/her careful reading of our manuscript. As stated in the revised version of the manuscript, the disproportionality analysis was performed using Microsoft® Excel® 2019 Version 2210. However, as indicated in the new subsection ‘Statistical Analysis’ (in Materials and Methods), statistical analyses for both quantitative and qualitative variables were performed using GraphPad Prism version 8.0.2. We hope that these modifications will meet the expectations of Reviewer 2.

  •     The figures caption should be placed below the figure.

We modified the revised manuscript accordingly.

Reviewer 3 Report

Dear authors

Thank you for asking me to review this manuscript. It reads very well, the method section is clear, even for a non-statistician as me and the results are presented sound and reasonable. I have only very minor suggestions to the discussion. I lack a section on implications for practice and research. Do you find that we lack knowledge about withdrawals and risk of abuse? Should we taper patients more carefully off these drugs? Should we be more restrictive in off-label use? More careful with treatment during pregnancy?

Author Response

Thank you for asking me to review this manuscript. It reads very well, the method section is clear, even for a non-statistician as me and the results are presented sound and reasonable.

We thank Reviewer 3 for these compliments and his/her understanding of our scope.

I have only very minor suggestions to the discussion. I lack a section on implications for practice and research. Do you find that we lack knowledge about withdrawals and risk of abuse? Should we taper patients more carefully off these drugs? Should we be more restrictive in off-label use? More careful with treatment during pregnancy?

We thank Reviewer 3 for helping us to clarify this point. Thus, we added the following paragraph to the discussion: ‘Regarding implications for practice and research, different areas have to be taken into account. During pregnancy, as recommended by the safety agencies [32–35,48–51], each antipsychotic treatment must be introduced or maintained following a careful assessment of the benefit/risk ratio. In children, off-label prescription should be properly substantiated, considering the scarcity of data regarding safety in efficacy in this population [50,51]. In adolescents, the risk of antipsychotic abuse and its characteristics should be further investigated. In addition, particular heed should be paid to specialized social networks, underlying new consumption patterns [14,16].’
